# Metabolic Profiles of Cancer Stem Cells and Normal Stem Cells and Their Therapeutic Significance

**DOI:** 10.3390/cells12232686

**Published:** 2023-11-22

**Authors:** Ioannis Stouras, Maria Vasileiou, Panagiotis F. Kanatas, Eleni Tziona, Christina Tsianava, Stamatis Theocharis

**Affiliations:** 1First Department of Pathology, Medical School, National and Kapodistrian University of Athens, 15772 Athens, Greece; stamtheo@med.uoa.gr; 2Section of Hematology and Medical Oncology, Department of Clinical Therapeutics, General Hospital Alexandra, 11528 Athens, Greece; 3Department of Pharmacy, School of Health Sciences, National and Kapodistrian University of Athens, 15771 Athens, Greece; 4School of Medicine, National and Kapodistrian University of Athens, 11527 Athens, Greece; pkanatas@outlook.com; 5School of Medicine, Aristotle University of Thessaloniki, 54124 Thessaloniki, Greece; helentziona@gmail.com; 6Department of Pharmacy, School of Health Sciences, University of Patras, 26504 Rion, Greece; tsianabachristina@gmail.com

**Keywords:** cancerstem cells, stem cells, metabolism, glycolysis, oxidative phosphorylation, lipid metabolism, amino acid metabolism

## Abstract

Cancer stem cells (CSCs) are a rare cancer cell population, responsible for the facilitation, progression, and resistance of tumors to therapeutic interventions. This subset of cancer cells with stemness and tumorigenic properties is organized in niches within the tumor microenvironment (TME) and presents altered regulation in a variety of metabolic pathways, including glycolysis, oxidative phosphorylation (OXPHOS), as well as lipid, amino acid, and iron metabolism. CSCs exhibit similarities as well as differences when comparedto normal stem cells, but also possess the ability of metabolic plasticity. In this review, we summarize the metabolic characteristics of normal, non-cancerous stem cells and CSCs. We also highlight the significance and implications of interventions targeting CSC metabolism to potentially achieve more robust clinical responses in the future.

## 1. Introduction

The tumor microenvironment (TME) has been receiving significant attention over the past years due to its pivotal role in the initiation, maintenance, and growth of the tumor bulk. Apart from the tumor cells, the TME consists of the extracellular matrix (ECM) and a variety of other cellular components, such as immune cells (e.g., natural killer cells, dendritic cells, CD4 and CD8 cells, tumor-associated macrophages, myeloid-derived suppressor cells, tumor-associated neutrophils, regulatory B- and T-cells) and stromal cells like cancer-associated fibroblasts (CAFs), cancer-associated adipocytes, endothelial cells, and pericytes. Each component of the TME, rather than exhibiting autonomous functions, closely interacts with the others via cell-to-cell contact or through the secretion of a diverse cytokine repertoire [1].

Cancer stem cells (CSCs) constitute a small population of cancer cells thought to be capable of initiating and promoting tumor growth and to play a crucial role in metastasis [2]. They are organized in niches within the TME, namely, specific anatomical areas that preserve the fundamental characteristics of CSCs, shield them from the immune response, safeguard their metabolic adaptability, which in turn allows for phenotype flexibility, and ultimately enhance their metastatic potential [3]. The TME can provide growth factors, cytokines, and extracellular matrix components that support CSC growth [4]. This crosstalk between TME and CSCs is mostly mediated by cancer-associated fibroblasts (CAFs) and cancer-associated endothelial cells [5,6]. Additionally, CSCs can evade the immune responseby enteringdormancy, a process, known as immune evasion, contributing to their resistance to conventional treatments. By expressing immune checkpoint proteins that contribute to an immunosuppressive microenvironment, CSCs can circumvent the immune system. Proteins such as PD-L1, which interacts with PD-1 on T-cells, lead to T-cell exhaustion and immune evasion. Additionally, CSCs secrete immunosuppressive factors, such as TGF-b, IL-6, and CCL20, and recruit immunosuppressive non-cancerous cells, such as regulatory T-cells and myeloid-derived suppressor cells, to support this immunosuppressive milieu. The inhibition of effector immune cells such as cytotoxic T-cells and natural killer cells (NK cells) promotes the survival and expansion of CSCs, making them less susceptible to immunotherapy [7].

CSCs can self-renew and differentiate similarly to normal stem cells. Unlike normal stem cells, CSCs are defined by their functional properties, whereas normal stem cells are defined by their location and differentiation potential. Additionally, CSCs display unique phenotypic markers that can be used to distinguish them from their normal counterparts. CSCs often exhibit a more complex and variable profile, characterized by the expression of specific surface markers. For instance, in breast cancer, CSCs are commonly identified by the overexpression of surface markers such as CD24, CD29 (β1-intergrin), CD44 (and its variants), CD49f, CD61, CD70, CD90, and CD133. Apart from these molecules, breast CSCs express non-CD surface markers such as CXCR4, epCAM, LGRS, Proc-R and intracellular markers such as SOX2 and are characterized by the overactivation of signaling pathways such as Wnt/β-catenin [2,4,8]. Additionally, CSCs frequently show increased resistance to standard treatments due to the overexpression of drug efflux transporters like ABCG2, which can serve as another “hallmark” of CSCs [7]. These are a few examples of the different phenotypic markers between CSCs and their healthy counterparts that demonstrate how crucial targeting CSC-specific markers is for theeffective and selective identification and isolation of CSCs.

The comparison between CSCs and normal stem cells extends beyond marker expression. These twocell populations employ different metabolic pathways to cover their complex bioenergy needs of self-renewal and differentiation. However, CSCs present altered regulation in a variety of metabolic pathways, including glycolysis, oxidative phosphorylation (OXPHOS), and lipid, amino acid, and iron metabolism. Although CSCs and normal stem cells present both metabolic similarities and differences, CSCs also possess the ability of metabolic plasticity, which provides them with unique adaptation ability.

Understanding the metabolic similarities and discrepancies between CSCs and their normal stem cell counterparts is of paramount importance in elucidating the unique properties of CSCs and their pivotal role in tumor progression. The disparities in metabolic pathway utilization—such as increased glycolysis, altered mitochondrial function, and heightened reliance on specific nutrient sources—reveal the potential for developing targeted therapies that exploit these vulnerabilities in CSCs while sparing normal stem cells. This review aims to summarize the metabolic profiles of CSCs and normal stem cells, offering important insights for the development of novel cancer therapies.

## 2. Metabolism of Normal Stem Cells

Normal stem cells utilize all available metabolic pathways to be able to both differentiate into specific cell types and retain a self-renewing reservoir. These include glycolysis, mitochondrial metabolism, and amino acid and lipid metabolism and are finely regulated according to the contextual needs of their environment.

### 2.1. Glycolysis

Glycolysis plays an integral role in the energy metabolism of pluripotent stem cells, generating a net of two ATP and two reduced NADH molecules per glucose molecule. Although it is much less efficient than the oxidative route, it has been observed that glycolysis is often preferred over the latter processas a catabolic pathway, as it can quickly yield ATP molecules and does not require the presence of oxygen; this is particularly important in hypoxic niches where stem cells reside during quiescence [9,10,11,12]. Glycolysis also shunts intermediates into the pentose phosphate pathway (PPP), producing substrates for nucleotide synthesis [13].

### 2.2. Mitochondrial Metabolism

Mitochondria are highly active organelles that majorly contribute to cellular metabolism, not only through ATP production, but also via cellular signaling, calcium homeostasis, and the biosynthesis of several macromolecules, such as fatty acids, amino acids, nucleotides, and hormones [14]. Mitochondrial structure and activity vary among different stages of cell pluripotency, as primed embryonic stem cells (ESCs) contain more elongated and less active mitochondria than their naïve counterparts [15,16]. Although overall smaller and functionally more immature as compared to the ones of somatic cells, mitochondria are believed to be integral mediators in establishing pluripotency, differentiation, and priming, partly due to their membrane potential [17,18]. Mice models have shown that a higher mitochondrial membrane potential increases the capacity of ESCs to differentiate into all three germ layers [19].

The TCA cycle is one of the main mitochondrial metabolic pathways, connecting glycolysis with oxidative phosphorylation (OXPHOS) and lipid and amino acid anabolism. Its main substrate, pyruvate, can serve many purposes, including producing lactateas a result of its reduction, generating acetyl-CoA and oxaloacetate, or fueling the TCA cycle to create intermediate substrates and supply the electron transport chain with active electron donors [20]. OXPHOS, on the other hand, is the most efficient way of ATP production, generating 32 ATP molecules per glucose molecule [21]. Simultaneously, the electron transport chain (ETC) contributes to reactive oxygen species (ROS) production, mitochondrial membrane potential, and protein import [22,23].

### 2.3. Lipid Metabolism

There is a finely regulated balance between anabolic and catabolic pathways of lipid metabolism in stem cells. Fatty acid oxidation (FAO) generates acetyl-CoA and NADH, supplying the TCA cycle and the ETC, respectively, and is regulated by the carnitine palmitoyl transferase (CPT) system. On the other hand, de novo lipid biosynthesis requires multiple substrates, including acetyl-CoA, TCA intermediates, and ATP, and is primarily mediated by fatty acid synthase (FASN) [24]. Multiple studies [25,26] suggest that lipids drastically affect the survival, maturation, differentiation, and proliferation of naïve stem cells through membrane biosynthesis [27], cellular signaling [28,29], and acylation of proteins involved in differentiation [30]. Yanes et al. showed that ESCs have a lipid profile different from that of differentiated cells and accumulate unsaturated fatty acids during the quiescent phase to maintain pluripotency [31]. Inhibition of FASN resulted in decreased reprogramming efficiency and apoptosis [32], while culturing PSCs in lipid-free E8 medium induced an intermediate naïve-to-primed pluripotent state due to an increased demand in de novo lipogenesis [33,34]. Lastly, lipogenesis is necessary for the induction and/or maintenance of pluripotency, because, apart from supporting the cells structurally, it also promotes mitochondrial fission, regulated by the enzyme ACC1 [35].

### 2.4. Amino Acid Metabolism

Normal stem cells rely on the intricate amino acid metabolism to maintain their capacity for self-renewal and differentiation. Amino acids play multifaceted roles in maintaining the delicate balance of stem cell functions. For instance, methionine metabolism, through its involvement in DNA biosynthesis, contributes significantly to the self-renewal of normal stem cells [36,37]. The regulation of tryptophan metabolism is critical for the regulation of vital stem genes and signaling pathways, which ensures the survival and self-renewal of these cells [36,38]. Lysine metabolism aids in reducing ROS and activating essential pathways such as Wnt, supporting self-renewal processes [39].

Threonine metabolism contributes to several processes that are essential for stem cell maintenance, such as energy production, acetylation, methylation, and nucleotide biosynthesis. Threonine dehydrogenase (TDH) is responsible for converting threonine to acetyl-CoA, which enters the TCA cycle for energy production and can be used for fatty acid synthesis [39,40,41,42,43]. Additionally, glycine generated from threonine metabolism contributes to one-carbon metabolism, facilitating nucleotide synthesis and serving as a precursor for methionine production. The glycine cleavage system (GCS) is a metabolic pathway that is highly activated in pluripotent stem cells during somatic cell reprogramming, and its activation promotes stem cell pluripotency by preventing cellular senescence and promoting histone H3 lysine 4 trimethylation (H3K4me3) modification through the expression of glycine decarboxylase (Gldc), a rate-limiting GCS enzyme regulated by SRY-box transcription factor 2 (Sox2) and Lin-28 homolog A (Lin28A) [44,45,46].

Furthermore, serine and glycine interconversion also plays a role in the one-carbon metabolism network, which is essential for protein, lipid, and nucleic acid synthesis and methylation reactions. Moreover, these processes affect the redox balance, influence ROS regulation, and maintain the NAD+/NADH and NADP+/NADPH ratios, which are vital for stem cell function and cellular homeostasis [47]. Serine metabolism in stem cells declines with age, reducing SAM availability for DNA methylation, which contributes to stem cell senescence [48].

In stem cells, valine is metabolized through a series of enzymatic reactions, including transamination and decarboxylation, yielding crucial metabolic intermediates, such as propionyl-CoA and acetyl-CoA [36]. These intermediates serve as substrates for various biosynthetic pathways, including fatty acid synthesis and energy production, thereby affecting stem cell growth and maintenance. Additionally, valine metabolism can influence the redox balance and contribute to the regulation of ROS, which is essential for cellular homeostasis and stem cell fate decisions [49].

Additionally, cysteine plays a crucial role in redox homeostasis and is a major component of the antioxidant glutathione. This amino acid is a rate-limiting precursor of glutathione biosynthesis, which is important for protecting cells from oxidative stress and maintaining their viability. Cysteine is catabolized via two main pathways. One pathway leads to the generation of pyruvate and α-ketoglutarate, which enter the tricarboxylic acid (TCA) cycle and contribute to energy production and biosynthesis. Additionally, enzymes involved in cysteine catabolism produce organic compounds that serve as carbon sources, further affecting carbon and energy metabolism in stem cells. These metabolic pathways play a pivotal role in regulating redox balance and ROS levels, thereby influencing stem cell pluripotency and functionality [50,51].

Proline plays a central role in stem cell functions and works as a direct scavenger of ROS. Glutamine is a critical energy source for ATP production, contributes to protein synthesis, and acts as an intracellular pH buffer. In addition, it is integral to the production of glutathione and supports the one-carbon metabolism network, which is vital for nucleic acid and methylation reactions. The multifaceted role of glutamine underscores its importance in maintaining the pluripotency and functionality of stem cells [52].

Finally, phenylalanine metabolism is also important to stem cells, as it is converted into tyrosine through enzymatic processes [53]. Eventually, tyrosine metabolism plays a critical role, with distinctive patterns of activation and downstream metabolite production. Enzymes involved in tyrosine metabolism, such as tyrosine aminotransferase (TAT), are essential for cellular processes in stem cells. TAT is responsible for the conversion of tyrosine into various downstream metabolites. One important role of TAT is to catalyze the conversion of tyrosine into 4-hydroxyphenylpyruvate, a key intermediate in the tyrosine catabolic pathway, which can have downstream effects on energy metabolism, neurotransmitter synthesis, and the regulation of oxidative stress [54].

### 2.5. Regulation of Normal Stem Cell Metabolism

Through quiescence, adult stem cells remain in G0 to maintain their stemness. During this stage, they reside in low-oxygen microenvironments that induce significant metabolic adaptations, essential to retain a slow-cycling circle—the most important of which is the expression of hypoxia-induced factor 1a (HIF-1a) [39]. HIF-1a induces the expression of PDK2 and PDK4 kinases that inhibit pyruvate dehydrogenase (PDH) and consequently mitochondrial metabolic activity, hence promoting anaerobic glycolysis as the main energy source for the cell [55]. Additionally, mitochondrial uncoupling protein 2 (UCP2) also hampers mitochondrial metabolism and promotes glycolysis and PPP [56], while the expression of lactate dehydrogenase A (LDHA) and hexokinase 2 (HK2) also contributes to maintaining pools of self-renewing hematopoietic stem cells (HSCs) [20].

Following stimulation, quiescent stem cells migrate to oxygen-rich microenvironments where the levels of HIF-1a, LDHA, HK2, and UCP2 are lower, and pyruvate starts fueling the TCA cycle and increasing mitochondrial respiration. One of the determining changes that also contributes to this metabolic switch is the PK gene splicing from PKM2 to PKM1, promoting OXPHOS activation [57,58,59]. Proliferative cells rely much more on the oxidative route for energy production, which also produces aspartate as a substrate for nucleotides [60].

Another characteristic of the proliferative phase is the marked increase in ROS production, as a consequence of OXPHOS activation [61]. Studies in animal models have shown that ROS are actually a prerequisite for stem cell maturation into terminally differentiated cells [62]. The targets of ROS in human stem cells remain unclear; however, there are several reports pointing towards prostaglandin E2, p38 MAPK, and other molecules, depending on the tissue [63]. Prostaglandin E2, which is a product of ROS-induced lipid oxidation in PSCs, promotes proliferation through Wnt signaling [64], whereas p38 MAPK activates IMPDH2, which in turn increases purine synthesis and proliferation inhuman HSCs [65].

### 2.6. Stem Cell Survival and Nuclear Reprogramming

As mentioned above, the metabolic profile of adult stem cells is mainly characterized byglycolysis. Glycolysis allows stem cells to survive in hypoxic microenvironments and also drives the nuclear reprogramming of somatic cells to induced pluripotent SCs (iPSCs) [66]. Moreover, glycolysis-derived acetyl-CoA affects pluripotency, as it is used for histone acetylation and stem cell renewal. Based on these observations, LIN28A/B was recognized as a possible important stemness factor, as it regulates glucose metabolism, histone methylation, and the metabolic proteome and influences one-carbon and nucleotide metabolism in mouse PSCs [67], while directly binding transcripts encoding for glycolytic and mitochondrial enzymes in human embryonic stem cells [68]. Another critical factor to be taken into consideration is that the inactivation of mitochondrial oxidation in favor of glycolysis protects stem cells from ROS; high levels of ROS effectuate genomic senescence, while low levels activate DNArepair mechanisms [69], mediate differentiation [70], and preserve the long-term self-renewal potential [71].

Lastly, amino acid metabolism also contributes to sustaining pluripotency through multiple pathways. Threonine and methionine donate methyl groups for anabolism and DNA methylation through one-carbon-metabolism [72,73], while glycine cleavage, induced by KLF4 and c-MYC, is essential for metabolic remodeling [46].

## 3. Metabolism of Cancer Stem Cells

### 3.1. Glycolysis

The work of Otto Warburg on cancer cell metabolism back in the dawn of the 20th century revealed the capability of cancer cells to cover metabolic needs via the pathway of glycolysis, even in environments with adequate oxygen supplies [74]. This form of aerobic glycolysis is known as the Warburg effect and holds great biomedical significance, constituting the principle of metabolicallyactive lesion detection via positron emission tomography (PET) imaging [75]. An evolving body of evidence supports the hypothesis that glycolysis not only serves as a key metabolic pathway in CSCs, but also can be utilized at even higher rates than in cancer cells [76]. The increased rate of glucose uptake by CSCs, the overexpression of genes coding for enzymes of the glycolytic machinery, as well as the enhanced production of ATP and lactate constitute evidence that supports the aforementioned statement [77]. Additionally, glucose depletion seems to be associated with compromised viability of CSCs in vitro [77], while environments rich in glucose enrich the tumor with CSCs [77]. Glycolysis has been observed in glioblastoma [78], osteosarcoma [79], breast [80], ovarian [81], lung [77], and colon CSCs [82]. Although glycolysis is generally less efficient in producing ATP than OXPHOS, it provides CSCs with significant metabolic intermediates that are utilized in a plethora of metabolic pathways, the most important of which is PPP, in order to meet complex bioenergy needs [83]. For instance, glucose-6-phosphate (G6P), produced from glucose phosphorylation by glucokinase, is channeled into PPP to generate NADPH [84] or ribose groups essential for nucleotide synthesis [85]. Similarly to what happens in iPSCs [86], glycolysis also plays a pivotal role in the adoption of a pluripotent phenotype by CSCs, as supported by evidence from breast [87], nasopharyngeal [88], and hepatocellular carcinomas [89]. Several different regulatory mechanisms were identified as playing a role in controlling glycolysis in CSCs. Firstly, the finding that MYC orchestrates CSC glycolysis aligns with its role in iPSCs [86]. Secondly, CD44, a stemness marker that binds to hyaluronic acid (HA) and mediates adherence to the extracellular matrix (ECM) [90], also participates in controlling glucose metabolism in CSCs [91]. Last but not least, glucose itself has the ability to induce the upregulation of glycolytic enzymes, such as hexokinases 1 and 2 (HK1, HK2), pyruvate dehydrogenase kinase 1 (PDK1), as well as the GLUT1 transporter, regulating the insulin-independent glucose influx [77]. Glycolysis upregulation was shown to convey resistance to radiotherapy in nasopharyngeal CSCs [88] and to chemoembolization in hepatocellular CSCs [92], possibly due to the enhanced feedback of PPP with G6P and the increased levels of NADPH that are able to ameliorate the detrimental effects of ROS produced by radiation.

### 3.2. Mitochondrial Metabolism

While the glycolytic metabolic phenotype offers a survival benefit in glucose-rich environments, providing CSCs with the ability to cover their proliferative energy needs, their localization within the tumor core does not always provide glycolysis-favoring conditions. By adopting a slow-cycling state of quiescence, CSCs are able to employ the OXPHOS machinery to cover their energy needs [93]. This is especially prominent in the paradigms of glioblastoma [94], leukemia [95], lung cancer [96], and pancreatic ductal adenocarcinoma (PDAC) [97]. The utilization of OXPHOS by CSCs is supported by their high mitochondrial mass and negative mitochondrial transmembrane potential, along with high rates of oxygen consumption and ROS generation [94,95,97,98,99,100,101]. Importantly, the adoption of the OXPHOS phenotype is involved to a great extent in the resistance to chemotherapy and targeted therapies that some CSCs possess. CSCs resistant to KRAS-targeting therapy in the PDAC context, as well as leukemic stem cells (LSCs) resistant to BCR-ABL-targeting in chronic myeloid leukemia (CML), were found to express an OXPHOS phenotype [102,103]. The most important regulator controlling mitochondrial biogenesis, an essential feature for the preservation of OXPHOS, is peroxisome proliferator-activated receptor-gamma coactivator-1alpha (PGC-1a) [104]. Besides PGC-1a, other regulators of the OXPHOS phenotype have been identified. In LSCs, the increased OXPHOS activity is under the influence of the adrenomedullin–calcitonin receptor-like receptor axis and spleen tyrosine kinase [105,106]. Another important regulator is nuclear factor erythroid 2-related factor 2 (NRF2), which was associated with a high mitochondrial potential, leading to more efficient OXPHOS, and was shown to play a role in the survival and treatment resistance that CSCs exhibit [107,108,109].

### 3.3. Metabolic Plasticity

The acquisition of either a glycolytic or an oxidative CSC metabolic phenotype is not exclusive. Instead, an increasing number of studies over the past years shed light on the metabolic plasticity that CSCs possess, which works towards the adoption of an intermediate glycolytic/OXPHOS phenotype with the ability to switch between the two. One of the most important pieces of evidence supporting this is the ability of breast CSCs to switch to OXPHOS when glycolysis is inhibited [87]. The precise control of the enzymes involved in glycolysis, OXPHOS, and the TCA cycle with the ultimate goal to generate ATP and preserve the NAD+/NADH ratio leads to a great metabolic plasticity [84]. NAD has been given significant attention as a factor controlling CSC plasticity [110], and there is evidence suggesting that the PGC-1a/MYC balance can affect CSC metabolic fate [97]. It is important to point out that the population of CSCs within a tumor is able to express a heterogenous phenotype, with different metabolic pathways expressed in different CSC subpopulations [111]. For example, epithelial-like breast CSCs adopt a different metabolic phenotype from mesenchymal-like ones, providing a survival benefit when it comes to adaptation in the presence of stressors [112].

### 3.4. Lipid Metabolism

Besides glycolysis and OXPHOS, lipid metabolism is also drawing attention in the context of CSC metabolism [113]. Mitochondrial FAO was shown to contribute to the survival and proliferation of CSCs by mitigating the oxidative load through the production of NADPH [114]. FAO also provides essential metabolic intermediates such as acetyl-CoA and NADH that favor ATP production [115]. Breast CSCs [116], LSCs [117], as well as normal HSCs [118] employ FAO to cover their bioenergy needs [119]. Carnitine palmitoyl transferase Ib (CPT1b), catalyzing the localization of long-chain fatty acids in the mitochondria for their subsequent beta oxidation [114], was found upregulated in breast CSCs under the influence of the JAK/STAT3 pathway, and this effect was mediated by leptin produced by the adjacent adipose tissue [116], implicating that TME components can modulate CSC metabolism. CPT-1b was shown to enhance stemness and chemoresistance in breast CSCs [116]. NANOG, a marker of cell stemness, promotes the overexpression of genes coding for enzymes of the FAO pathway [89]. The inhibition of FAO was shown to specifically tackle the CSC population [116]. Besides FAO, fatty acid synthesis was observed in pancreatic CSCs [120]. Lipases participating in fatty acid synthesis, such as ATP citrate lyase (ACLY), acetyl-CoA carboxylase (ACC), and fatty acid synthase (FASN), controlled by the lipogenic transcription factor SREBP1c, were found to be upregulated in CSCs [121,122]. Furthermore, the levels and the phosphorylation of AMPK in CSCs were lower, leading to enhanced lipase activity and an increase in the levels of malonyl-CoA, which serves as a fatty acid synthesis precursor [123]. Fatty acid storage in lipid droplets (LDs) is essential for their management, and an increased LD content was described in colorectal CSCs [124]. Additionally, cholesterol synthesis via the mevalonate pathway greatly participates in CSC generation [125]. Cholesterol is incorporated in membrane lipid rafts, ensuring the smooth operation of signaling pathways essential for CSC proliferation [115]. Finally, lipidomics showed that fatty acid desaturation, which promotes membrane fluidity, contributes to stemness preservation in ovarian and glioblastoma CSCs [126,127], and the regulation of this process by stearoyl-CoA desaturase (SCD1), NF-κB, and aldehyde dehydrogenase I family member A1 (ALDH1A1) promotes stemness in colorectal CSCs [128,129].

### 3.5. Amino Acid Metabolism

Amino acids can serve as another energy source in CSC metabolism. Glutamine is a key amino acid that is converted by glutaminase to glutamate, which undergoes deamination to alpha-ketoglutarate (aKG) [130]. aKG generation, along with pyruvate and oxaloacetate, can promote anaplerosis of the TCA cycle in order to balance the constant loss of citrate to mitochondria for lipid synthesis [131]. Glutamine also provides the essential carbon and amino nitrogen for the generation of other lipids, nucleotides, and amino acids [132] and regulates major epigenetic modifications [133]. It was shown that pancreatic CSCs exhibit a strong preference for glutamine [134], and glutamine metabolism was also implicated in the ability of colorectal CSCs to attach to liver tissue in vitro [135].

Iron metabolism dysregulation is recognized as another CSC metabolic trait [136]. Generally, iron abundance in CSCs enhances stemness [137,138]. Iron supplementation was shown to promote stemness in lung carcinoma, breast carcinoma, and cholangiocarcinoma cell lines, whereas iron chelators ameliorated this effect [138,139,140]. Additionally, in non-small cell lung cancer (NSCLC) CSCs, iron enhanced tumor invasion via hydroxyl radicals [141]. CSCs seem to have greater iron needs than cancer cells. In comparison to cancer cells, CSCs showed lower expression of ferroportin 1 (FPN1) and hephaestin, which regulate iron outflow, and higher levels of transferrin receptor (TfR), which controls the iron influx through TF/2Fe^3+^. Similar to glycolysis, the CD44 stemness marker promotes iron accumulation by interacting with TfR [142,143]. Table 1 providesan overview of the similarities and differences between CSC and normal stem cell metabolism.

## 4. Targeting CSC Metabolism

### 4.1. Glycolysis

As mentioned in Section 3, glycolysis was found to play a pivotal role in CSC metabolism, and targeting this key metabolic pathway is expected to have a major impact on CSC populations. The molecule 2-deoxyglucose (2-DG) acts as a glycolysis inhibitor by blocking the formation of G6P and inhibiting HK2 and phosphoglucoisomerase [144]. Human CD44+/CD23-low breast CSCs were found to undergo apoptosis after 2-DG treatment and exhibited higher sensitivity to doxorubicin, a chemotherapy agent serving as the backbone of breast cancer therapeutics [80]. In the context of pancreatic cancer, the inhibition of HK2 by 3-bromopyruvate (3-BrPA) sensitized pancreatic CSCs to gemcitabine [145] and mitigated the in vitro viability and in vivo tumorigenicity of CSCs [77]. In glioblastoma CSCs, pentyl-3-bromopyruvate ester, another agent blocking glycolysis at the level of HK2, reduced cell stemness, as reflected by the downregulation of CD133, and its coadministration with doxorubicin mitigated the cancer-initiating properties of glioblastoma CSCs in an in vivo murine model [146]. Dichloroacetic acid (DCA) blocks pyruvate dehydrogenase kinase (PDK) and subsequently enhances pyruvate dehydrogenase (PDH) to feed the TCA cycle with acetyl-CoA from pyruvate. DCA was shown to increase the oxygen consumption rate (OCR) and radiosensitivity in brain CSCs by reducing LDH activity and to enhance the action of etoposide [147]. DCA also favors the dimerization of the PKM2/OCT4 complex, resulting in lower transcriptional activity of OCT4 and the induction of apoptosis in glioma CSCs [147]. Additionally, vitamin C blocks glycolysis [148] and inhibits the USP28/MYC/SLUG axis by reducing lactate formation through epinephrine-induced LDHA blockade [149]. Finally, the inhibition of glucose uptake through GLUT1 blockade by the small molecule WZB117 appeared to exert a significant action on ovarian, glioma, and pancreatic CSCs [150].

### 4.2. Mitochondrial Metabolism

Concerning mitochondrial metabolism, the evolutionary theory which supports the hypothesis that eukaryotic mitochondria originated from aerobic bacteria, has promotedthe utilization of antibiotics to target mitochondrial metabolism [151]. This appears to apply to CSC metabolism. Antibiotics such as doxycycline and azithromycin inhibit CSC sphere-forming ability [152]. Tigecycline, an antibiotic targeting mitochondrial ribosomes, was used in combination with the BCR-ABL tyrosine kinase inhibitor imatinib and was found to be effective against CML LSCs in vitro as well as in an in vivo animal model [103]. Apart from mitochondrial ribosomes, electron transport complexes constitute an appealing target when it comes to tackling the CSC population. Metformin is a widely used antidiabetic agent that blocks complex I and induces a state of energy crisis in CSCs, with profound effects on CD133+ pancreatic cells [153]. Since metformin uptake is dependent on organic anion transporters, phenformin, a more hydrophobic biguanide derivative, could achieve a more effective mitochondrial localization [154]. Preclinical data support its efficacy in NSCLC by inhibiting complex I [155]. Salinomycin, another drug blocking OXPHOS, was identified after a screen of breast CSCs possessing an EMT phenotype [156]. Treatment with salinomycin was found to dampen stemness in in vivo studies [156] and affect colorectal CSCs in spheroids mimicking TME conditions [157]. Verteporfin, a Food and Drug Administration (FDA)-approved agent for the treatment of macular degeneration, blocks complexes III and IV and was shown to specifically affect glioblastoma CSCs [158]. The inhibition of H+-ATP synthase by oligomycin dampened the ability of CD87+ lung CSCs to form spheres [159]. Other OXPHOS inhibitors like antimycin and rotenone were able to selectively target CD44+/CD117+ CSCs, sparing CD44+/CD117- non-CSCs [159]. The inhibition of complex I and II enzymes utilizing flavin by diphenyleneiodonium chloride (DPI) was also shown to negatively impact CSC subpopulations [160]. A similar effect was observed with tri-phenyl phosphonium (TPP), the action of which is highly specific for CSC mitochondria due to their high transmembrane potential [161]. Furthermore, compounds functionally inhibiting CSC activity known as “mitoketoscins” structurally resemble coenzyme A and block the OXCT1 and ACAT1 catalytic regions found in its binding site [162]. mDIVI1 is a molecule targeting DRP1, a key protein inmitochondrial division, that was documented to affect CSC signaling [163]. Drug resistance constitutes a major obstacle when assessing agents against a single aspect of mitochondrial metabolism, such as rotenone. Instead, drugs with multiple “hits” on OXPHOS such as menadione, which not only inhibits complex I, but also increases the generation of mitochondrial ROS, are a more favorable choice to overcome resistance [154]. It was shown that increasing ROS to viability-threatening levels could be a better way to target CSCs at the level of mitochondrial metabolism [97].

### 4.3. Lipid Metabolism

Lipid metabolism was documented to play a pivotal role in multiple key CSC features discussed in Section 3. The inhibition of lipogenesis by targeting key enzymes of this metabolic process has profound effects in managing CSC populations. Cerulenin, a FASN inhibitor, ameliorated stemness in brain CSCs, as shown by the downregulation of stemness markers like Sox2 and CD133 [164]. Another FASN inhibitor, resveratrol, exerted its action against CD44+/ESA+/CD23- ovarian CSCs in vivo [165]. At the level of acetyl-CoA carboxylase (ACC), there is evidence that soraphen A, a documented ACC inhibitor, has a pronounced effect on breast CSCs [166]. Lipid desaturation was also described as an attractive target, given the significance this process holds for maintaining cancer cell stemness. SCD1 inhibition by CAY10566 restricted cancer cell stemness [127]. Finally, targeting FAO limited the available energy sources necessary for CSC growth and maintenance. Evidence from both in vitro and in vivo investigations on CPT1 inhibitors such as etomoxir and perhexiline shows their efficacy against breast CSCs [167]. Etomoxir was also able to increase the sensitivity of hepatocellular carcinoma (HCC) CSCs to sorafenib [88] and showed synergy with ABT-737, a molecule targeting BCL-2, in the context of acute myeloid leukemia (AML) [168]. The more selective FAO inhibitor, avocatin B, was shown to tackle CSC subpopulations and spare normal HSCs [169].

### 4.4. Amino Acid Metabolism

In the context of amino acid metabolism, targeting glutamine metabolism has been the common denominator of the efforts on tackling CSCs. Compounds like 968 and BPTES reduce pluripotency in HCC CSCs by blocking glutaminase and affecting the Wnt/β-catenin signaling pathway. Another glutaminase inhibitor, zaprinast, was able to enhance the effects of radiotherapy on pancreatic CSCs and promote apoptosis via an increase in ROS to non-viable levels [170]. Figure 1, created using https://biorender.com (accessed on 16 October 2023), summarizes some of the therapeutic interventions shown to target CSC metabolic pathways.

## 5. Limitations

The study of CSC biology is subject to several limitations. Isolation protocols and CSC characterization processes lack consistency [154]. A common ground needs to be established to overcome the discrepancies that may arise and to ensure both the validity and the reproducibility of the generated data. Additionally, isolating CSCs from cell lines is largely inadequate, since they lack the diversity of their physiologic state and cannot faithfully replicate the TME conditions. Thus, the conduction of future studies involving promptly obtained patient-derived stem cells or stem cells with a short passage history is required [154]. In this case, the challenge of extracting and culturing CSCs from primary tumors needs to be considered, as it hinges on marker specificity [136]. The definition and the examination of CSCs through the scope of functional attributes can overcome the scarcity of surface markers. A commonly employed method of studying CSCs in their natural habitat involves the generation of 3D models that closely mimic the CSC niche [171]. In the context of CSC therapeutics, CSC metabolic plasticity, which allows CSCs to switch to more favorable metabolic processes in response to stressors, limits the effectiveness of single targeting. Instead, the inhibition of at least two metabolic pathways is expected to deliver a more robust “hit” on CSC subpopulations. Given the heterogeneity of the CSC population within a tumor, it is crucial to identify the dominant pathways within a particular CSC group, since this would enable the utilization of more targeted agents and maximize their efficacy [84].

Furthermore, since normal stem cells and CSCs share a variety of metabolic pathways, targeting CSC metabolism may raise concerns on systemic adverse effects secondary to the dysregulation of normal stem cell metabolism. Therefore, the conjugation of CSC metabolism-targeting agents with antibodies (ADCs) targeting CSC surface markers or their encapsulation in exosomes specifically directed towards CSCs may lower the risk of non-specific targeting.

## 6. Conclusions

It is evident that CSCs constitute a TME subpopulation of paramount importance in modulating tumor growth, recurrence, and metastasis. Understanding their metabolic profiles and addressing the pathways they utilize to cover their complex bioenergy needs—as well as the similarities and differences with respect to the ones employed by normal stem cells—is key in shaping more reliable identification methods and effective therapeutic interventions that will ensure more robust and longer clinical responses. This is the cornerstone of the era of theranostics, where the elucidation of the mechanisms that discriminate CSCs from their normal counterparts could be exploited to reveal CSC vulnerabilities and shed light on novel therapeutic targets.

## Figures and Tables

**Figure 1 cells-12-02686-f001:**
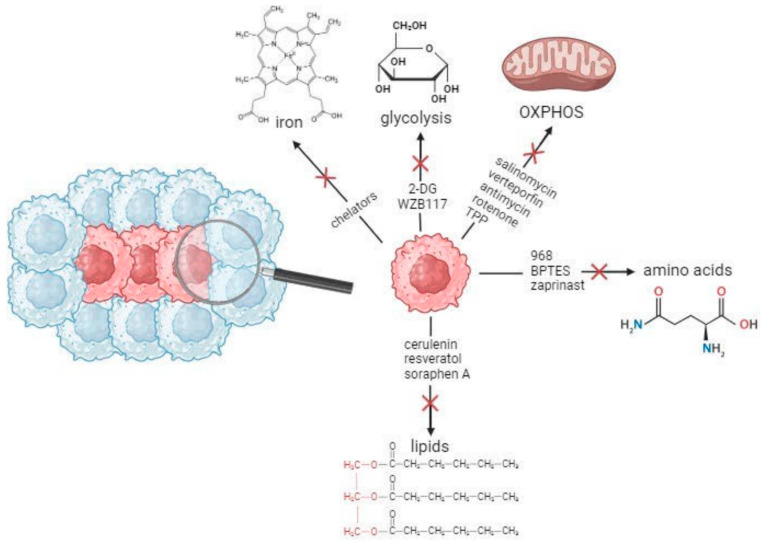
Overview of some of the therapeutic agents targeting CSC metabolic pathways. CSCs, shown in red, employ distinct metabolic pathways to cover their energy needs, namely, glycolysis, OXPHOS, lipid, amino acid, and iron metabolism. Chemical agents and small molecule inhibitors included in the arrows have shown promising results at the preclinical level.

**Table 1 cells-12-02686-t001:** Summary of the similarities and differences between CSC and normal stem cell metabolism.

Metabolic Pathway	CSCs	Normal Stem Cells
Glycolysis	promotion of pluripotency	
	regulation by MYC	
	feeds PPP with intermediates for nucleotide synthesis	
	promotion of radioresistance	metabolic advantage in hypoxic niche
	enhanced by CD44	
OXPHOS	metabolic flexibility in glucose-deprived conditions	high membrane potential favors ESC differentiation
	promotion of chemoresistance	
	regulation by PGC-1a	
Lipid Metabolism	lipid desaturation promotes stemness	
	FAO provides NADPH and ATP	fatty acid synthesis induces and maintains pluripotency
	increased lipid droplet content	
	enhanced cholesterol synthesis	
Amino Acid Metabolism	glutamine promotes glutathione production and ROS control	
	lysine catabolism favors liver metastasis in colon CSCs	methionine enhances self-renewal
		lysine catabolism mitigates ROS levels
		tryptophan regulates stem gene expression
		threonine provides energy
		glycine prevents senescence
		serine supports one-carbon metabolism network
		valine maintains cellular homeostasis
		cysteine promotes glutathione production and ROS control
		proline serves as a ROS scavenger
Iron Metabolism	intracellular abundance promotes stemness and invasion	

## Data Availability

Not applicable.

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
