# Peer review of "Metabolic Profiles of Cancer Stem Cells and Normal Stem Cells and Their Therapeutic Significance"

_cells, 2023, doi:10.3390/cells12232686_

Round 1

Reviewer 1 Report

Comments and Suggestions for Authors

This is an interesting and well-written review focusing on cancer stem cell (CSC) metabolism and its therapeutic implications.

The review describes the main metabolic pathways activated in both normal and tumor stem cells and presents data supporting the possible eradication of CSCs by targeting specifically activated metabolic pathways.

The authors provide a clear and comprehensive picture of CSC metabolism, I only have minor points to comment on.

·       1. In the introduction, I would recommend moving the initial part describing TME after the introduction about CSC. Furthermore, since the review is focused on the parallelism between CSCs and normal stem cells, I would emphasize and go more deeply into the commonalities and differences regarding the niche, properties and markers used for their identification.

·       2.  Chapter 2.2 is entitled Regulation, what does it mean?

·       3.  Table 1 is not of easy to understand, could you modify the graphics of the table to highlight similarities and differences between normal and CSC?

·         4.Limitations: could the authors comment on the limitation of targeting metabolic pathways with regard to normal stem cells or systemic effects?

Comments on the Quality of English Language

The English language is mostly fine

Reviewer 2 Report

Comments and Suggestions for Authors

The manuscript by Stouras et al. compares metabolic properties of normal- and cancer stem cells (CSC) as well as summarizes therapeutic targets in regard to metabolic profile of CSC. Overall, it is an elegant summary of current knowledge about metabolic characteristics of CSC in comparison to normal stem cells. 

I find the Table 1 very confusing. The table must be properly formatted in order to make clear cut between the described metabolic pathway.

The authors aim to compare metabolism of normal stem cells and CSC. Targeting metabolic pathways in CSC seems to be very important in regard to cancer treatment. Authors should discuss whether targeting CSC can affect normal stem cells as they share some similarities.
